# Performance, Modeling, and Cost Analysis of Chemical Coagulation-Assisted Solar Powered Electrocoagulation Treatment System for Pharmaceutical Wastewater

**Tharaa M. Al-Zghoul [1,\*], Zakaria Al-Qodah [2,\*] and Ahmad Al-Jamrah [1]**

[1] Department of Civil Engineering, School of Engineering, University of Jordan, Amman 11942, Jordan
[2] Department of Chemical Engineering, Faculty of Engineering Technology, Al-Balqa Applied University, Amman 11134, Jordan
\* Correspondence: tharaaalzghoul@gmail.com (T.M.A.-Z.); zak@bau.edu.jo (Z.A.-Q.)

**Abstract:** The combination of the chemical coagulation-assisted electrocoagulation (CC-EC) process with a solar photovoltaic energy source has attracted increasing attention for the efficient removal of chemical oxygen demand (COD) from pharmaceutical wastewater. In this paper, the CC-EC process has been utilized as an alternative to conventional chemical processes for the treatment of pharmaceutical wastewater. The effects of the various operating parameters, such as coagulant dosage, coagulant type, number of electrodes, the distance between electrodes, electrode configuration, operating time, and current density, on COD removal efficiency were investigated. The results indicated that the optimum conditions were achieved at 500 mg/L of alum dosage, 3.105 mA/cm$^2$ of current density, six electrodes with a distance of 4 cm between electrodes, and the MP-S electrode configuration, where the operating cost of conventional energy was 0.283 \$/m$^3$. Indeed, by using the CC process alone, the COD removal efficiency was 26% and 61.5% at the optimal dosages of 750 mg/L of NaOH and 500 mg/L of alum, respectively. In the CC-EC treatment, the removal efficiencies of COD were 88.7, 92.9, 94.4, and 89.4% using six electrodes, 2 cm of distance between electrodes, MP-S electrode configuration, and 20 min with 1.553 mA/cm$^2$ of current density, respectively. The removal efficiencies of COD achieved through CC, EC, and CC-EC processes were 61.5, 85.4, and 94.4%, respectively.

**Keywords:** chemical coagulation; electrocoagulation; pharmaceutical wastewater; combined treatment processes; solar-powered treatment systems

## 1. Introduction

The world has recently witnessed a growing population, the acceleration of urbanization, and the impact of climate change, which put pressure on water resources, causing them to become limited resources in terms of both quality and quantity [1]. Although the need for clean water is a critical issue in developing countries, contemporary countries in addition suffer from a permanent shortage of clean water resources due to pollution from urbanization and industrial processes [2]. In particular, the Mediterranean region is one of the most vulnerable regions around the world to climate change, population growth, and limited water resources [1]. Jordan, as one of the Mediterranean countries, is regarded as the world's second-poorest country in terms of water resources due to high temperatures and a high population growth rate, and it has been classified as a semi-arid to arid country due to its dependence primarily on rainwater [3–5]. The high demand for water in various industries, such as chemicals, pharmaceuticals, paper, textiles, and printing, results in environmental destruction and ecosystem threats due to the discharge of highly toxic and non-biodegradable wastewater [6]. One of the major concerns of the twenty-first

century is reducing the impacts of global water shortages [7–9]. Among all industrial activities, the pharmaceutical sector releases relatively high quantities of waste because of the significant increase in the number of factories and production capacity and the increasing demand for medication and thus wastewater production. As a result, large-scale pharmaceutical wastewater (PhWW) discharge has become a source of worry owing to the substantial contamination of water bodies and chronic or sub-chronic toxicity to aquatic ecosystems as well as humans via accumulation in the environment and food chain [10,11].

Pharmaceutical effluent differs from the rest of the conventional effluents in terms of its discharges of organic pollutants and drug components, which have unknown environmental consequences and are becoming one of the biggest challenges for the environment. Pharmaceutical wastewater contains a high concentration of stubborn organic pollutants, high levels of wasted solvents, significant concentrations of many inorganic salts, different kinds of pharmaceutical residue, high levels of total dissolved solids (TDS), biochemical oxygen demand (BOD), chemical oxygen demand (COD), and suspended solids (SS) [12,13]. As a result, it discharges large quantities of wastewater rich in various types of pollutants such as salts, acids, and alcohol, as well as containing large amounts of chemical oxygen demand (COD). When wastewater containing a significant level of COD is discharged into streaming water, it lowers the dissolved oxygen and contaminates the water [14]. Therefore, it is necessary to treat this type of wastewater before discharging it to avoid many environmental and health risks [15–18].

Given the current global concern about water scarcity, sustainable water resource use should be one of the primary goals of many industries, particularly those with high water consumption [9]. The tendency to reclaim treated wastewater, especially in arid and semi-arid areas, plays a fundamental and vital role in the sustainable management of water resources, particularly in terms of reusing it in diverse applications such as industry, agriculture, indirect human uses, etc. [19,20]. The process of wastewater reclamation is one of the best solutions to the problem of water scarcity [21]. However, the treatment of pharmaceutical wastewater is a very challenging task and faces many constraints in terms of finding an appropriate treatment technology due to its intractable behaviors and the cost of treatment related to its energy consumption. Thus, various conventional treatment techniques are applied to overcome the problems of wastewater, including advanced oxidation processes, catalytic oxidation [22], photo catalysis [23], ion exchange [24,25], reverse osmosis [26], biological processes [27–29], and adsorption [30,31]. However, due to their shortcomings, such as inefficiency, limited biodegradability, low COD and BOD removal efficiencies, and high cost, none of these processes met the required standards [32,33]. Therefore, there is a need to find a cost-effective, high-pollutant removal efficiency, and eco-friendly treatment process [31].

One of the best processes used for wastewater treatment is the electrocoagulation (EC) process, where the principle of its work is based on electrically dissolving either iron or aluminum ions from iron or aluminum electrodes and using them as coagulants in the formation of ions. At the anode, metal ions are produced, and at the cathode, hydrogen gas is released. The flocculated particles floated out to the water surface by the hydrogen gas bubbles formed on the cathode [34–36]. The arrangement of the electrodes might be either monopolar or bipolar. The potential advantages of EC, such as its flexibility, simple operation, no addition of chemicals, short startup period, ease of control, and ability to deal with different pollutants, have increased interest in its implementation [27,37–40]. Additionally, it is possible to run the EC process using renewable energy sources, such as solar panels, fuel cells, and windmills [41]. However, using the EC procedure as a single treatment step might impose significant practical limitations, such as electrode passivation, particularly if the wastewater is highly loaded with COD. Electrode passivation can be minimized by reducing the concentration of COD in EC feed wastewater. This can be achieved by diluting the raw wastewater with treated wastewater or by using a pretreatment process, such as the chemical coagulation process (CC). However, wastewater

dilution will increase its volume, and this imposes a larger size for treatment units and hence increases both capital and operational costs [26]. Accordingly, the use of a pre-treatment step in the EC process is a more efficient and cost-effective alternative.

One more parameter that affects the applicability of the EC process is energy consumption. It is clear from the literature that most electrocoagulation research has relied on conventional electrical power sources, which are notorious for their high-energy consumption, lengthy treatment durations, and excessive sludge production. In some regions, such as remote towns with limited access to the energy grid, a wastewater treatment system is difficult to power with a conventional electric power source [42]. Photovoltaic (PV) energy has many advantages, including low fossil fuel consumption, low carbon dioxide emissions, free usage, a long service life, low maintenance costs, and no pollution [43,44]. This research focused on the capability of a solar PV energy source to eliminate COD from pharmaceutical wastewater. The employment of the EC method as a single treatment method might impose significant practical limits, particularly if the wastewater is extremely contaminated; thus, there is a need for treatment methods that are effective. As a result, using the EC in conjunction with a pre or post-treatment method will improve its performance [40–46]. Several studies have been conducted regarding the combination of EC with other technologies, such as biological treatments and chemical coagulation (CC) [47–49]. Combining techniques can improve the treatment performance while potentially saving energy and potential costs. Combining CC and EC processes can optimize treatment by shortening separation times and reducing generated sludge water content as well as streamlining the sludge dewatering process [47–49].

As mentioned above, pharmaceutical wastewater is a very complex effluent containing high loads and complex chemical pollutants. Based on previous reviews, it is concluded that combined treatment systems are more efficient than single processes for achieving high removal efficiencies of most types of pollutants. For this reason, the objective of this research is to investigate the utility and feasibility of a suitable treatment system for pharmaceutical wastewater. According to some preliminary results in our lab, a combination of CC and EC processes, which have never been applied before to such wastewater, could achieve better results than those achieved by single-treatment methods. In order to reduce the treatment operation cost, a solar-powered combined CC-EC system will be applied to achieve high COD removal efficiencies and obtain treated wastewater that fully meets the Jordanian Standards and Metrology Organization (JSMO) requirements. These requirements are 500 and 100 ppm, for COD and BOD, respectively. The treated wastewater will be suitable for reuse in many applications, including irrigation and agriculture limits. This solar-powered combined treatment system, which has rarely been used before, is expected to be efficient for such industrial wastewater. Firstly, chemical coagulation will be applied to the raw wastewater to remove colloidal and suspended particles. Then, pretreated wastewater by the CC process will be fed to the EC to eliminate most of the remaining COD.

## 2. Materials and Methods

### 2.1. Pharmaceutical Wastewater

The wastewater employed in this study was obtained from the wastewater treatment plant of one of the pharmaceutical companies in Amman, Jordan. Wastewater samples were collected in clean plastic containers from the inlet to the treatment unit with a capacity of 20 L, totaling roughly 100 L, and transported to the laboratory located in the Al Balqa Applied University—Faculty of Engineering and Technology, Amman. Several preliminary tests were conducted to investigate the physical and chemical characteristics of pharmaceutical wastewater, including pH, temperature, conductivity, $BOD_5$, and initial COD. The remainder was kept in the refrigerator at 4 °C to be used as needed. This was one of the objectives of the study. Table 1 lists the initial characteristics of the pharmaceutical wastewater sample used in this study.

**Table 1.** Initial characteristics of pharmaceutical wastewater.

| Parameter | Value |
|---|---|
| pH | 6.32 |
| Conductivity (mS/cm) | 8.31 |
| T (°C) | 29 |
| Initial COD (mg/L) | 3447.9 |
| $BOD_5$ (mg/L) | 930.9 |

*2.2. The Combined Treatment System*

As mentioned, the treatment system in our study consists of two subsequent processes, which are: (1) CC and (2) EC. The reason for this application is to reduce the organic load with the CC step in order to obtain the high efficiency of the EC step.

2.2.1. Chemical Coagulation (CC)

The laboratory scale consisted of a cylindrical plastic reactor (3000 mL) with dimensions of (height of 18 cm and diameter of 16 cm) and a stirrer (Stuart Scientific Stirrer SS3, UK) set at 1700 rpm for 6 min, which was followed by slow mixing at 150 rpm for 4 min. In this study, alum ($Al_2(SO_4)_3 \cdot 18H_2O$) was used as a coagulant and caustic soda (NaOH) was used as a softening agent at different concentrations (250, 500, 750, and 1000 mg/L). For each coagulant (when using a 500 mg/L coagulant dosage, for example), 1.5 g of coagulant was dissolved in 5 mL of distilled water and then added to 3000 mL of wastewater sample for 10 min to provide a homogeneous solution in the reactor. In all chemical coagulation experiments, the samples were allowed to settle for 30 min; then, 100 mL samples were taken, filtered using a paper filter, and placed in a glass bottle for a COD test of the treated wastewater. Finally, after the CC process, the samples were filtered and placed in the EC reactor.

All wastewater contains nitrate ions. The procedure for COD measurement is standard for this type of wastewater. The COD test was carried out by mixing sulfuric acid ($H_2SO_4$), silver sulfate ($Ag_2SO_4$) reagent and potassium dichromate oxidizing agent ($K_2Cr_2O_7$) with the sample to be analyzed during this test. The mixture was boiled and refluxed for two hours at 150 °C to achieve a maximum oxidation. The sample was allowed to cool at room temperature, and the amount of potassium dichromate left for COD determination was measured by calibrating the spectrophotometer with a blank sample at a zero reading. Inserting standard 10 mL ampoules into the spectrophotometer (HACH DRB200, Tokyo, Japan) immediately determines the COD of the sample. COD is the amount of oxygen consumed in the oxidation reaction (in mg/L).

For the BOD test, the sample is prepared, and the measurement reagent is estimated to determine the sample size. The appropriate volume of the sample is selected and placed in the BOD container along with the magnetic stirrer rod. To prevent nitrification, a few drops of a nitrification inhibitor (ATH) are usually added to the sample bottle, plus a few drops of 3–4 drops of potassium hydroxide solution are placed in the sealing gasket to absorb the carbon dioxide ($CO_2$), and then, the sealing gasket is inserted into the neck of the bottle. The BOD sensors are installed in the sample bottle and inserted into the shelf of the bottle. According to the instructions for BOD (HACH DRB200, Japan), the measurement begins by incubating the sample for 5 days and counting the temperature of 20 °C.

The chemicals used in this study are of analytical grade. These include NaCl, $H_2SO_4$, HCl (Hiba medicals), $Al_2(SO_4)_3 \cdot 18H_2O$ (Philip Harris), NaOH (POCH), HCl (SDFCL), $Ag_2SO_4$ (Riedel-de Haën), $K_2Cr_2O_7$ (Riedel-de Haën), KOH (GCC), and $CO_2$ (JGC).

### 2.2.2. Solar-Powered Electrocoagulation (SPEC)

The batch EC reactor used in this study was a rectangular, plastic reactor with a volume of 3 L and dimensions of 29 cm × 8 cm × 13 cm. Figure 1 shows the schematic view of the solar-powered electrocoagulation process used in this study.

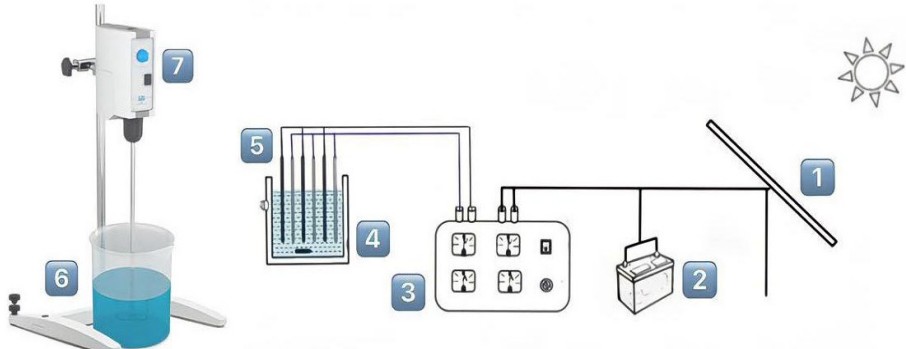

**Figure 1.** Schematic view of the SPEC process: 1. photovoltaic module, 2. battery, 3. charge controller, 4. EC reactor, 5. electrodes, 6. CC reactor, and 7. Starrier.

As shown in Figure 1, in the EC process, iron electrodes were utilized as sacrificial electrodes due to their low cost and ease of access. Six electrodes were dipped in a solution to a depth of 6 cm with a spacing of 4 cm between the electrodes. Plastic spaces were used to adjust the electrode and maintain the electrodes vertically parallel. Three electrodes were used as cathodes, while the remaining three electrodes were used as anodes. All the electrodes were rectangular with dimensions of (9 cm × 6 cm × 1 mm) with circular holes. The electrodes that were immersed in the pharmaceutical wastewater solution had a total effective surface area of approximately 644.08 cm². The source of power supply was a monocrystalline silicon photovoltaic panel (PS-M36S-90, Amman, Jordan) with a maximum power of 90 watts used with a charge controller and regulator, (10A 12V/24V 240W), Future Electronics, Cairo, Egypt. Normal wires of 1.5 mm were used to connect the electrodes in a monopolar parallel, and PVC was used to isolate them. A battery (NPP 12-5.0 12V5.0Ah/20HR, NNP, Guangzhou, China) was installed to store energy, and a Digital potentiometer voltmeter (Drok YB27VA, Drok, Guangzhou, China) to regulate the current.

As mentioned above, this study used iron (Fe) as an electrode material. According to the following chemical reactions, ferric hydroxide is formed according to chemical Equations (1)–(3), and then, it acts as a coagulant for the pollutants found in the wastewater [50]:

$$Anode: \mathrm{Fe_{(s)}} \leftrightarrow \mathrm{Fe^{2+}_{(aq)}} + 2\mathrm{e^-} \tag{1}$$

$$Cathode: 2\mathrm{H_2O_{(l)}} + 2\mathrm{e^-} \leftrightarrow \mathrm{H_{2(g)}} + 2\mathrm{OH^-_{(aq)}} \tag{2}$$

$$Overall: \mathrm{Fe_{(s)}} + 2\mathrm{H_2O_{(l)}} \leftrightarrow \mathrm{Fe(OH)_{2(s)}} + \mathrm{H_{2(g)}} \tag{3}$$

In each experiment, after the CC process, the treated wastewater samples were filtered and placed in the EC reactor. The EC process was carried out under different parameters, including (1.553–4.658 mA/cm²) current densities, (2–4 cm) inter-between distance electrodes, (1–3) pairs of electrodes, and (MP-P, MP-S, and BP-S) electrode configuration, with an operating time of (5–60) minutes with alum coagulants at different dosages. We add 1000 mg/L NaCl to be used in the EC process to obtain conductivities. During the EC process, an oxide is formed at the anode. After each experiment, the electrodes were rinsed with HCL (0.1 N) solution to remove any solid residues on the surface of the reactors and electrodes, and then, they were rinsed again with distilled water to avoid

electrode passivation. In addition, the used electrodes were recovered by polishing the surface oxide layer with abrasive paper, washed in HCl (0.1 N) solution, rinsed with distilled water, and then dried with absorptive paper and finally weighted. These values are utilized in the calculations of the total operating cost, and then, the samples are taken from the bottom of the reactor using a pipette and filtered for the COD test. These steps have been repeated for the rest of the operational parameters.

Factors Influencing the EC Process Efficiency in COD Removal

The batch EC reactor used in this study was a rectangular, plastic reactor with a volume of 3 L and dimensions of 29 cm × 8 cm × 13 cm. Figure 1 shows the schematic view of the solar-powered electrocoagulation (SPEC) process used in this study.

Several investigations into COD removal by the EC process have found that several factors influence the removal efficiency. Using a holistic approach, the influencing factors are divided into two groups using a holistic approach: (i) physicochemical solution properties, and (ii) operational parameters relating to reactor design [36]. The operating parameters that usually affect the performance of the EC process are shown in Figure 2.

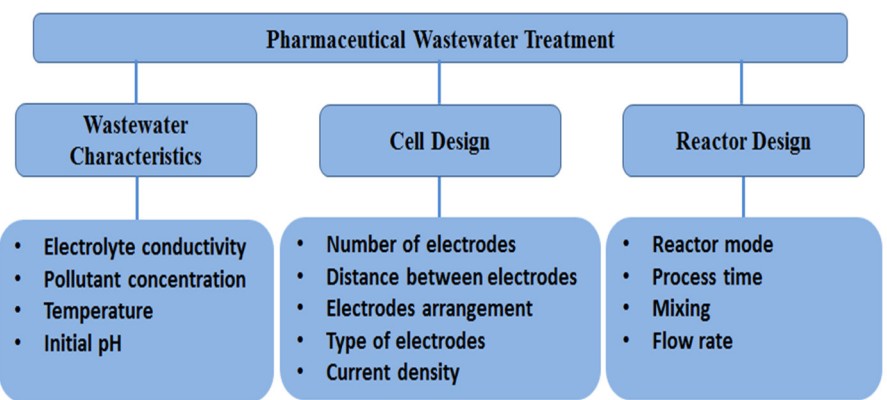

**Figure 2.** Parameters affecting the performance of the EC treatment process.

As shown in Figure 2, the parameters affecting the performance of the EC process include the wastewater properties such as pollutant concentration, pH, temperature, and conductivity. On the other hand, the operational parameters include the type, number arrangement of the electrodes and the distance between them, current density and reaction time.

*2.3. Analytical*

Before and after the CC and EC treatment processes, several quality parameters were examined to indicate the removal efficiency of COD in the treated wastewater samples. The COD was measured by the titration method. The COD value was reported in mg/L. A pH meter (WTW ProfiLine pH 3310 Meter, Xylem, Washington, DC, USA) was used to measure temperature and pH. Electrical conductivity was measured using (CON 6, LaMotte-Europe, Warwick, UK). The electrical conductivity value was reported in mS/cm. A standard analytical balance (THB-300 Scale, Italy) was used in this study for analytical procedures and to measure a change in electrode mass.

**3. Results**

Table 1 shows the characterization of pharmaceutical wastewater before any treatment. The BOD5 and COD concentrations in the pharmaceutical wastewater were 930.9 and 3447.9 mg/L, respectively. In addition, because of the low BOD5/COD ratio (0.27), such effluents are unsuitable for biological aerobic processes. As a result, the pharmaceutical effluent had to be treated before it could be used for irrigation, agriculture, etc.

### 3.1. Chemical Coagulation Pretreatment Process

Coagulation is a method based on the collision of charged particles in a colloidal suspension with counter-ions so that they are neutralized, agglomerated (the small particles are converted into large particles called flocs), and then precipitated [51]. The CC method was used before the EC method in this study, primarily to reduce the contaminant load entering the EC cells, thus improving the EC performance. Several authors have recently investigated the use of alum ($Al_2(SO_4)_3 \cdot 18H_2O$) and sodium hydroxide or caustic soda (NaOH) as softening agents for most wastewater treatment [52–54]. As a consequence, coagulant was added to pharmaceutical wastewater to promote particle instability and size growth, allowing the organic chemicals present as COD to be successfully removed. Coagulant dosage is an important factor in determining how metal ions react with organic matter in wastewater to improve its removal [53,55]. It was noted that with the addition of coagulants, the pH level of the solution started to increase up to 7. The results of the COD removal are presented in Figure 3. A summary of the experimental results under the effect of coagulant dosage on COD removal efficiency is listed in Table 2.

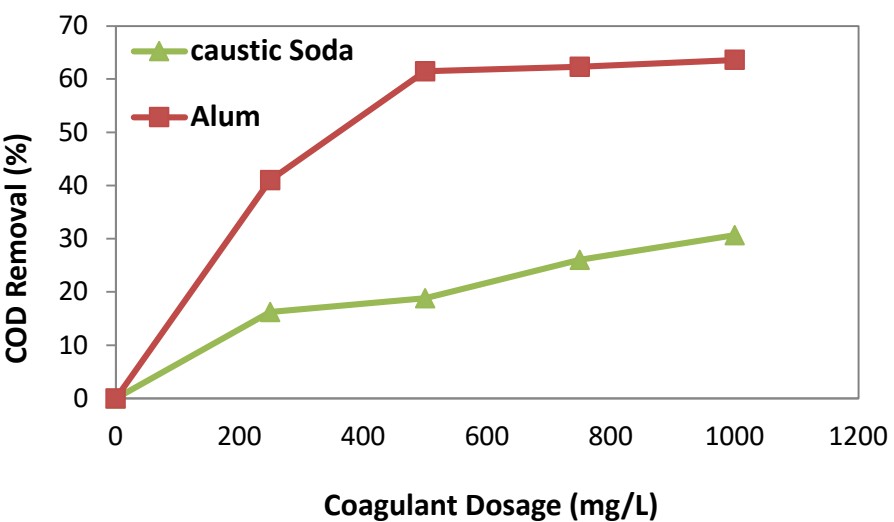

**Figure 3.** Effect of coagulant dosage on COD removal efficiency.

**Table 2.** Effect of different coagulants dosage on the removal efficiencies of CC treatment system.

| Coagulant Dosage (mg/L) | COD Removal (%) | |
| --- | --- | --- |
| | Caustic Soda (NaOH) | Alum ($Al_2(SO_4)_3 \cdot 18H_2O$) |
| 0 | 0 | 0 |
| 250 | 16.2 | 41 |
| 500 | 18.8 | 61.5 |
| 750 | 26 | 62.3 |
| 1000 | 30.7 | 63.6 |

According to Figure 3 and Table 2, the COD removal value was 26% at the optimal dosage of 750 mg/L of NaOH for COD removal, while the COD removal value was 61.5% at the optimum dosage of 500 mg/L of alum for COD removal. However, COD removal efficiencies were higher at 1000 mg/L alum and 1000 mg/L NaOH, with 63.6% and 30.7% COD removal, respectively. In addition, it was shown that the removal efficiency of COD increased slowly when the alum dosage was greater than 500 mg/L. This indicates that it is not necessary to add more alum than 500 mg/L. A similar trend was reported by Maleki [51] and Bouchareb [56], where alum had an advantage over other chemicals and showed the best removal efficiencies for COD. In general, increasing the coagulant dosage increased the efficiency of COD removal by coagulant.

The remaining chemicals showed relatively low COD removal efficiency and were ranked in the following order: alum, then caustic soda. The COD removal efficiencies when using 500 mg/L of chemical dosage were 61.6 and 18.9%, respectively. Nurul Hanira [53] obtained similar results in a previous study concerning the removal of ammonia from leachate (with caustic soda addition as a softener agent). A high COD removal efficiency was not obtained when using caustic soda. Caustic soda might be an inferior chemical and not very effective in removing COD. An extended settling period might be required to reduce the concentration of the precipitates that contributed to the COD removal [52,57].

The findings of the chemical coagulation process indicate that while the removal efficiency of COD from pharmaceutical wastewater is high, the concentration of contaminants in the CC process effluent does not meet the requirements of the Jordanian Standards and Metrology Organization (JSMO). As a result, another treatment process for pharmaceutical wastewater treatment must follow the CC process. For this reason, an SPEC process was applied as the second treatment step to further reduce the COD level in the remediated wastewater to values that meet the required standards.

### 3.2. Process Performance of SPEC

One of the most promising processes that has garnered the most attention from researchers recently is EC. When a current is applied, the anode oxidizes, while the cathode reduces in an aqueous solution. Fe electrodes are the most frequently used because of their different benefits, such as low cost and accessibility. In this method, coagulation or precipitate is formed in situ, such as with Fe hydroxides [58,59]. In the present study, the EC process was employed as a post-treatment method to further treat pharmaceutical wastewater. The use of the SPEC process was intended to treat COD efficiently and economically. Therefore, in this study, the effect of different operating parameters, including current density, reaction time, the distance between electrodes, electrode number, and electrode configuration, on the SPEC process of pharmaceutical wastewater was investigated. In addition, the operating costs (OC) of this process were evaluated.

### 3.2.1. The Effect of the Electrode Number

As a parameter for the treatment of pharmaceutical wastewater, the effect of the number of electrodes on the removal efficiency of COD was studied. In this study, different numbers of electrodes (two, four, and six electrodes) were used with the following operating conditions: a pH of 7, a distance between electrodes of 4 cm, an operating time of 20 min, and an MP-P electrode configuration. In Figure 4, the relationship between the electrode numbers and the COD removal efficiency of pharmaceutical wastewater after the EC process is shown.

Figure 4 depicts how the number of electrodes affects the removal efficiency of COD using the EC process. It can be seen that the more electrodes there are, the more efficient the removal of COD can be. In the treatment using six electrodes, the removal efficiency of the COD was 88.7%. The removal efficiency of the COD after processing using four electrodes was 88.4%. Meanwhile, the removal efficiency of the COD using two electrodes is 87.8%.

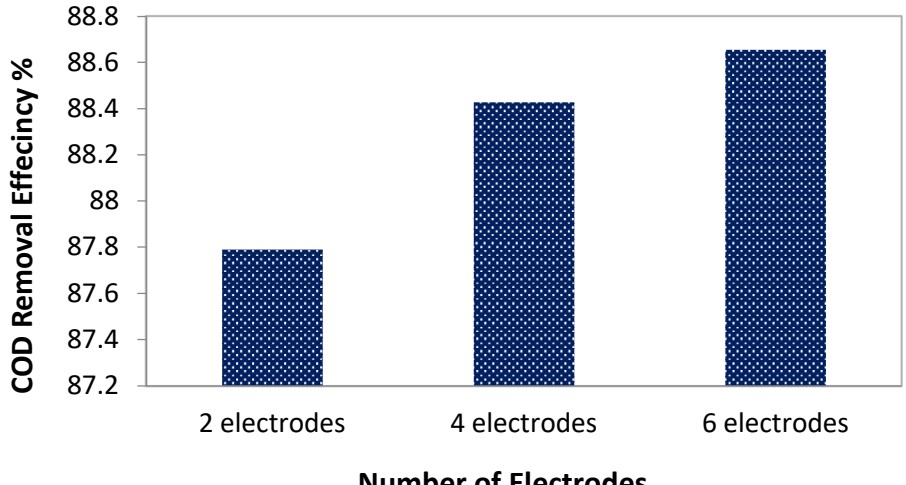

**Figure 4.** The effect of a number of electrodes on the EC process. Experimental conditions: The electrode material is Fe, the pH is 7; the volume of the sample is 3000 mL, the electrolysis time is 20 min, and the distance between the electrodes is 4 cm.

This is explained by the fact that because the area is larger and there are more electrodes being utilized, the current density will be lower than optimum. Furthermore, the production of $Fe^{3+}$ and $OH^-$ ions increased with the number of electrodes, which may have contributed to the production of $Fe(OH)_3$ as a coagulant. These outcomes are consistent with those attained by Gatsios [60] and Salih Muharam [61].

3.2.2. Effect of Distance between Electrodes

In the EC process, the effect of the distance between electrodes on the removal efficiency of COD was studied as a parameter for the treatment of pharmaceutical wastewater. The distance between the electrodes was varied between (2, 3, and 4 cm). The effect of the distance between electrodes on the COD removal efficiency by the EC process is shown in Figure 5.

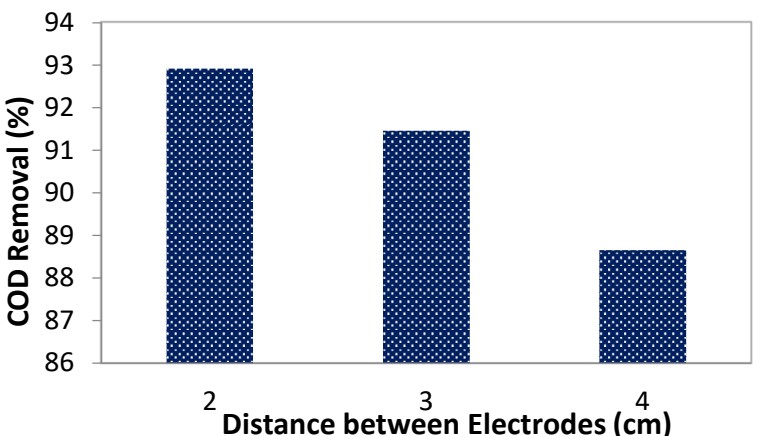

**Figure 5.** The effect of the distance between electrodes on the EC process. Experimental constant parameters: The electrode material is Fe; the temperature is 28 °C; the operating time is 20 min.

As shown in Figure 5, the removal efficiency of COD in the EC process decreases slightly when the distance between electrodes increases due to the slower rate of electron transfer. It was observed that the COD removal efficiency decreased (from 88.7 at 4 cm to 92.9 at 2 cm) the distance between electrodes. The potential (V) increases together with

the increase in distance between the electrodes. As a result, resistance increases and has a negative impact on pharmaceutical wastewater treatment. These outcomes are consistent with those attained by Salih Muharam [61], Janpoor [62], Nasrullah [63], and Bhagawan [64].

### 3.2.3. Effect of Electrode Configuration

Figure 6 shows the COD removal efficiencies of six electrodes with different electrode configurations (MP-S, MP-P, and BP-S).

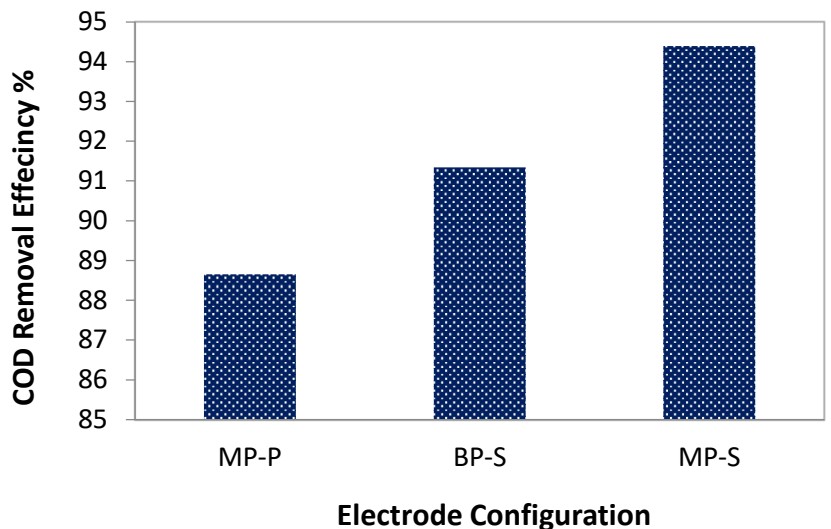

**Figure 6.** The effect of electrode configurations on COD removal efficiency in the EC process. The following conditions were used in the experiment: six electrodes, a distance of 4 cm between electrodes, a pH of 7, and a 20 min operating time.

Figure 6 shows the effect of electrode configuration on the removal efficiency of COD using the EC process. It was observed that the highest COD removal efficiency was with the MP-S configuration compared to the BP-S and MP-P configurations, which increased from 88.7% to 94.4%. The COD removal efficiency reached a maximum of 94.4% with the MP-S configuration. In a study conducted by Kobya [65] and Naje [66], it was discovered that the MP-S electrode connection mode in the EC process removed more arsenic than other electrode connection modes.

### 3.2.4. Effects of Reaction Time and Current Density

The reaction time and current density for COD removal from pharmaceutical wastewater have been investigated at different reaction time intervals (5–60 min) and three different current densities (1.553, 3.105, and 4.658 mA/cm²). Figure 7 shows the effects of the reaction time and current density on the COD removal efficiencies under the following operating conditions: six electrodes, 4 cm inter-distance electrodes, and an MP-P electrode configuration.

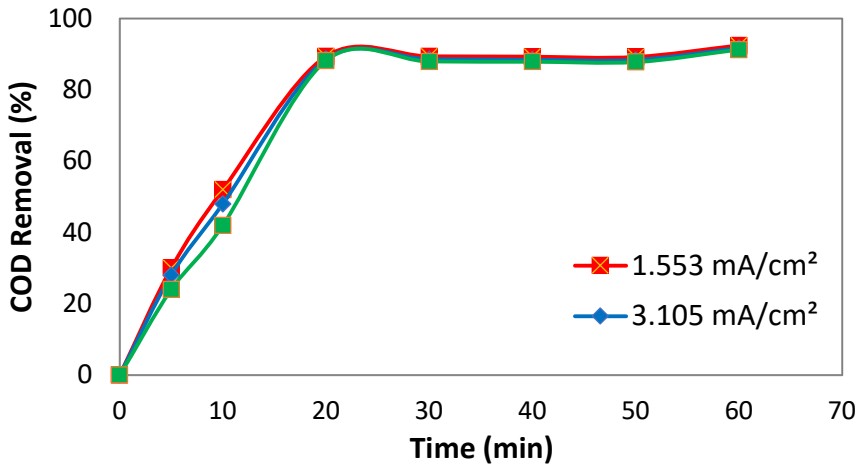

**Figure 7.** Effect of the reaction time and current density on EC for different current densities and the following experimental conditions: six electrodes, a distance of 4 cm between electrodes, and an MP-P electrode configuration.

As shown in Figure 7, the removal efficiency of COD increases at a relatively high rate during the first 20 min, then decreases, and reaches a maximum removal efficiency of COD after 60 min of reaction time. Figure 7's findings indicate that a reaction time of 20 min is sufficient for nearly complete COD treatment efficiency. After 20 min, treatment efficiency shows an insignificant improvement due to the passivation layer on the electrode material. Therefore, the optimum reaction time was 20 min when the removal efficiencies for each (1.553, 3.105, and 4.658 mA/cm$^2$) at 20 min reached about 89.4%, 88.7%, and 88.3%, respectively. These outcomes are consistent with those attained by Janpoor [62], Nasrullah [63], and Bhagawan [64]. The COD removal efficiency did not increase as the CD was increased, but it did decrease slightly when the CD was increased (from 1.553 to 4.658 mA/cm$^2$) at 20 min (from 89.4 to 88.3%), which could be attributed to the passivation layer on the electrode material.

A summary of the experimental results under the effect of different EC cell parameters is listed in Table 3. In addition, the effect of the operating times on COD removal efficiencies at different current densities is shown in Table 4.

**Table 3.** Effect of electrode arrangement, number, and distance on the removal efficiencies of EC and CC-EC treatment systems.

| Operating Parameters | | COD Removal (%) | |
|---|---|---|---|
| | | EC | CC-EC |
| Distance between Electrodes (cm) | 2 | 63.56 | 92.92 |
| | 3 | 54 | 91.46 |
| | 4 | 44.58 | 88.65 |
| Number of Electrodes | 2 | 68.28 | 87.79 |
| | 4 | 69.96 | 88.43 |
| | 6 | 70.53 | 88.65 |
| Electrode Arrangement | MP-S | 63.56 | 94.4 |
| | BP-S | 55 | 90.5 |
| | MP-P | 38.08 | 88.65 |

**Table 4.** COD removal efficiency for EC alone and the combination of chemical coagulation and electrocoagulation (CC-EC) as a current density (MP-P of the electrode configuration, six electrodes, and a distance between electrodes of 4 cm).

| Time (min) | CD (mA/cm²) | COD Removal (%) | |
| --- | --- | --- | --- |
| | | EC | CC-EC |
| 20 | 1.553 | 72.52 | 89.42 |
| | 3.105 | 70.53 | 88.65 |
| | 4.658 | 69.49 | 88.25 |
| 30 | 1.553 | 73.42 | 89.77 |
| | 3.105 | 71.45 | 89.01 |
| | 4.658 | 70.32 | 88.57 |
| 40 | 1.553 | 74.76 | 90.28 |
| | 3.105 | 72.76 | 89.51 |
| | 4.658 | 71.75 | 89.12 |
| 50 | 1.553 | 77 | 91.14 |
| | 3.105 | 74.87 | 90.32 |
| | 4.658 | 73.48 | 89.79 |
| 60 | 1.553 | 80.41 | 92.46 |
| | 3.105 | 78.49 | 91.72 |
| | 4.658 | 77.32 | 91.27 |

It is clear from Table 3 that the removal efficiency was maximum when the distance between the electrodes, the number of electrodes, and their arrangement were 2 cm, six electrodes, and MP-S, respectively. On the other hand, Table 4 shows that the removal efficiency decreases as the current density increases. This can be attributed to the fact that as the current density increases, the electrode temperature increases, which increases the rate of electrocoagulation in the first period, but fast precipitation on the electrodes causes what is called electrode passivation.

3.2.5. Kinetic Study

In this study, the kinetic study was performed based on COD removal efficiency from the pharmaceutical wastewater and evaluated at various current densities (1.553, 3.105, and 4.658 mA/cm²), a constant pharmaceutical wastewater volume of (3000 mL), six electrodes in an MP-P electrode configuration, 4 cm distance between electrodes, and ambient temperature. For such a batch solar photovoltaic EC method, the mass conservation of COD is [67]:

$$-\frac{dC_A}{dt} = -r_A = -kC_A \tag{4}$$

where $(-r_A)$ is the COD removal rate in mg/L/min and $t$ is the EC time in minutes. With the first-order reaction kinetic model $((-r_A) = k_1 C_A)$, the integration of Equation (4) at the initial concentration of $C(0) = C_0$ gives:

$$C_A = C_{A_0} e^{K_1 t} \tag{5}$$

where $k_1$ is the first-order reaction rate constant (in min⁻¹). Therefore, the linearization of Equation (5) can be given as:

$$\ln C_A = \ln C_{A_0} - k_1 t \tag{6}$$

For the second-order reaction kinetic model $((-r_A) = k_2 C_A^2)$, the integration of Equation (6) will lead to a time-dependent concentration being obtained as:

$$\frac{1}{C_A} = \frac{1}{C_{A_0}} + k_2 t \tag{7}$$

where $k_2$ is the reaction rate constant of the second order in (L/mg/min).

To estimate the time required for COD removal, the kinetics of the EC removal reaction of COD must be examined. The values of the rate constants and regression coefficients were determined by fitting the needed course performance data with first- and second-order kinetic equations and calculations as shown in Table 5, Figures 8 and 9. Here, $r_c$ is the rate of conversation, $t$ is the time, $C$ is the final COD in the solution, and $k_1$ and $k_2$ are the first-order and second-order rate constants in min$^{-1}$ (L/mg/min), respectively. In addition, the reaction rate coefficient and $R^2$ values for COD and CD are summarized in Table 5.

**Table 5.** Predicted parameters of first and second-order kinetic model COD removal efficiency at different CD with solution volume = 3000 mL, conductivity = 8.31 mS/cm, and pH = 7.

| Parameters | CD (mA/cm²) | First-Order Kinetic Model $k_1$ (min$^{-1}$) | $R^2$ (–) | Second-Order Kinetic Model $k_2$ (L/mg/min) | $R^2$ (–) |
|---|---|---|---|---|---|
| | 1.553 | $8.22 \times 10^{-3}$ | 0.9122 | $2.65 \times 10^{-5}$ | 0.8854 |
| COD | 3.105 | $7.66 \times 10^{-3}$ | 0.9052 | $2.25 \times 10^{-5}$ | 0.8792 |
| | 4.658 | $7.06 \times 10^{-3}$ | 0.8943 | $2.01 \times 10^{-5}$ | 0.8681 |

The first-order kinetic model can successfully simulate the removal efficiency of COD in the EC at various CD because the data are well correlated (higher $R^2$). According to Table 5, when the current density increases, the first-order kinetic constant decreases, and the maximum kinetic rate is 0.00822 (min$^{-1}$) at 1.553 mA/cm², showing the maximum removal efficiency.

Figures 8 and 9 show the kinetic data of the EC reaction obtained in this study.

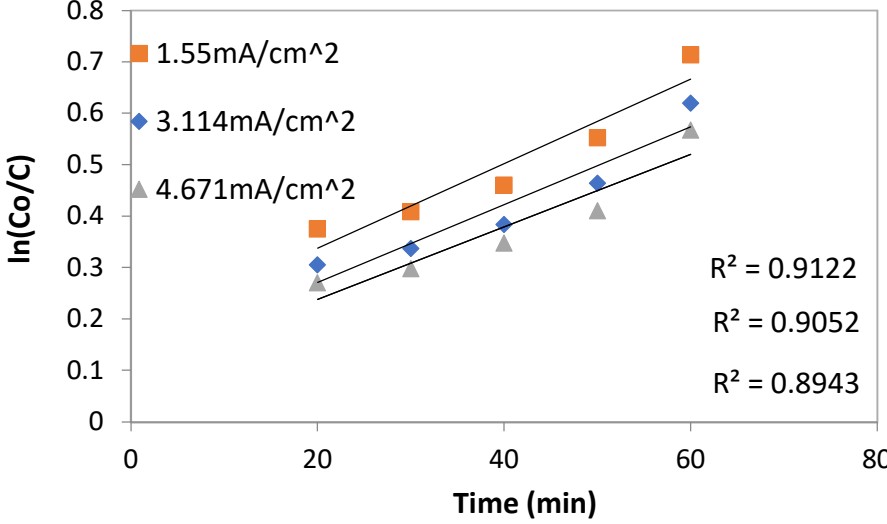

**Figure 8.** Kinetics study of EC at first-order reaction.

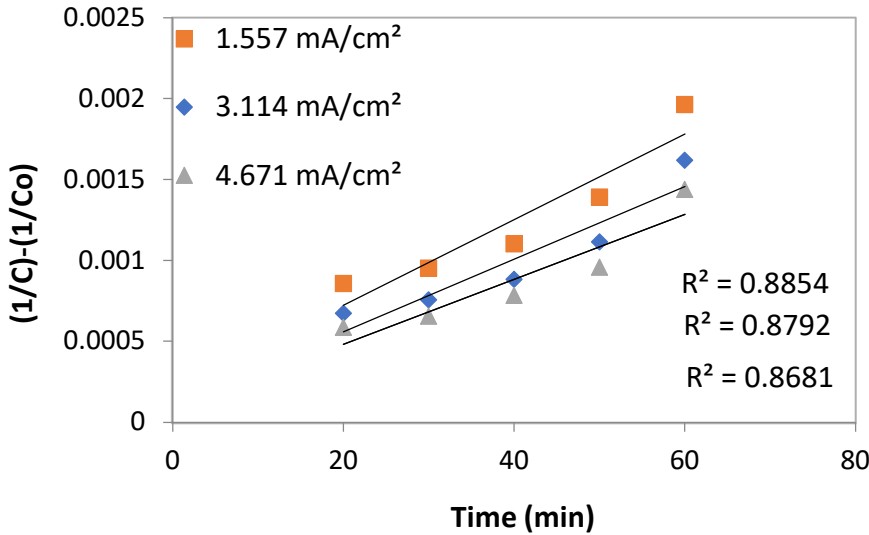

**Figure 9.** Kinetics study of EC at the second-order reaction.

As illustrated in Figures 8 and 9, the first-order kinetic model has a higher $R^2$ and therefore a more suitable regression coefficient than the second-order regression coefficient. The values of $R^2$ were calculated in the range of 0.8943–0.9122 and 0.8681–0.8854 for the first and second-order kinetic modules, respectively. The kinetic evaluation results of the treatment of pharmaceutical wastewater by the EC method are compatible with the results obtained by Ahmadian [68] in the treatment of slaughterhouse wastewater by the electrocoagulation method.

*3.3. Comparison of CC, EC, and Combined CC-EC*

The results were used to compare the performance of CC, EC, and combined CC-EC in COD removal from pharmaceutical wastewater under optimum conditions of 500 mg/L of alum dosage, 3.105 mA/cm² of current density, six electrodes with a distance of 2 cm between electrodes, and MP-S electrode configuration. On the other hand, the results of normal conditions of 250 mg/L of alum dosage, 1.553 mA/cm² of current density, four electrodes with a distance of 4 cm between electrodes, and MP-P electrode configuration are shown in Figure 10.

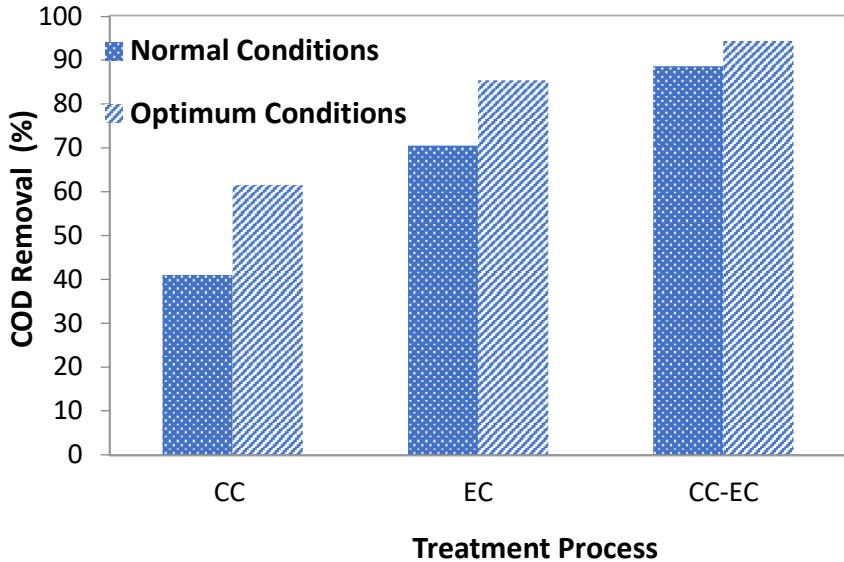

**Figure 10.** The overall removal efficiency of COD using the combined treatment systems.

Figure 10 shows the values of the removal efficiency of COD from pharmaceutical wastewater when applying a single treatment method such as CC and EC processes and a combined CC-EC process. These values, when operating under normal and optimum conditions, are 41% and 70.5% for CC, 61.5% and 85.4% for EC, and 88.7% and 94.4% for the CC-EC combined process. It can be observed that the combined CC-EC process achieved the highest COD removal efficiency, as the COD removal efficiency improved, reaching 94.4% at optimum conditions compared to the single CC and EC processes. The results shown in Figure 10 demonstrate that under optimum conditions, the combination of the CC and EC processes together boosts COD elimination efficiency by 9.5% as compared to the EC process alone. However, in the CC method, the combined CC-EC process indicated that the COD removal efficiency was greater than 34.9%. At this step, the treated wastewater can be used in agriculture, irrigation, and other industries, in addition to achieving the standards required by the JSMO. These outcomes are consistent with those attained by Swain [49], Salih Muharam [61], Can [69], and Al-Qodah [70], who reported that the combined CC-EC process showed the highest pollutant removal efficiency compared to the single CC and EC processes. These results confirm the success of this combined treatment process in removing COD without suffering from the problem of electrode passivation.

The literature shows that several combined treatment processes have been developed to treat pharmaceutical wastewater. Some of these combined processes achieved significant results in removing the pollutants found in this wastewater. Table 6 shows the most important results of these studies.

**Table 6.** Combination of several processes for pharmaceutical wastewater treatment. EF: electro-Fenton, PcO: photocatalytic oxidation, PF: photo-Fenton, AFFBR: anaerobic fixed film bed reactor, ICME: iron/carbon (Fe-C) micro-electrolysis, EGSB: expand granular sludge bed, MR: microalgae reactor, HSW: high-strength wastewater, and LSW: low-strength wastewater.

| Combined Treatment | Abbreviation | Operating Conditions | Removal Efficiency | Reference |
|---|---|---|---|---|
| Combination of electro-co-agulation (EC), electro-Fenton (EF) and photo-catalytic oxidation (PcO) | EC + EF | 1 h EF, 5 mA/cm$^2$ | 64% TOC | [71] |
| | EC + PcO | 4 h PcO, Fe:$H_2O_2$ molar ratio as 1:10 | 70.2% COD | |
| | EF + PcO | 1.5 g/L $TiO_2$ 10 mM $H_2O_2$ | 97.8% $BOD_5$ | |
| Solar-driven photo-Fenton (PF) followed by subsequent biological treatment | PF + biological | pH hydrogen peroxide dosage iron concentration applied voltage | 84% of COD for LSW 82% of COD for HSW | [72] |
| Ozone-based advanced oxidation and adsorption | AO-Ad | pH (5–11), 3 h | 75–88.5% COD | [73] |
| | | activated char for adsorption | 85.4–92.7% COD | |
| Combined electrocoagula-tion followed by anaerobic fixed film bed reactor (AFFBR) | EC-AFFBR | pH 7.2 80 A/m$^2$ of CD 25 min | 24% COD 35% BOD 70.25 of color removal | [74] |
| Hybrid coagulation, gamma irradiation, and bi-ological treatment | CC-GI | coagulants: $Ca(OH)_2$, $FeCl_3$ and $Al_2(SO_4)_3$ oxidants: gamma-rays, $H_2O_2$ and $S_2O_7^{-2}$ | (92.7% ± 2.3%) of COD for LSW (90.2% ± 2.9%) of COD for HSW | [75] |

Based on the findings summarized in Table 6, it can be noted that the combined CC-EC used in this study showed that COD removal was the highest compared with other combined processes. Therefore, the results further confirm that the CC-EC process may be a better option to treat pharmaceutical wastewater.

### 4. Operational Cost (OPC) Analysis for Solar Photovoltaic Electrocoagulation

This section describes the estimation of conventional and solar operating costs for the EC process and their connection to the reaction time. Equations (8) and (9) are utilized to calculate the electrode and energy consumption for the treatment of pharmaceutical wastewater.

$$ELC = 1000 \times \frac{I t_{EC} M_W}{ZFV} \tag{8}$$

where *ELC* is electrode consumption in kg/m³; *I* is the direct current in A; *m* is the specific amount of electrode material dissolved in kg/m³; $t_{EC}$ is electrocoagulation time in seconds; and *Z* is the chemical equivalence of the electrode (for Fe, $Z_{Fe}$ = 2). In addition, $M_w$ is the molecular weight of the electrode metal ($M_{w,Fe}$ = 56 g/mol); *F* is the Faradays constant (96,500 C/mol); and *V* is the volume of the treated pharmaceutical wastewater treatment in m³.

$$ENC = \frac{(m)(P)(I)(t_{EC})}{V} \tag{9}$$

where *ENC* is the specific electrical energy consumption (kWh/m³), *I* is a direct electrical current in A, *P* is the applied voltage in V, $t_{EC}$ is the EC time in hours, and *V* is the volume of the treated wastewater in L [76].

An economic analysis of the solar-powered EC process was performed to find out the total operating cost because it is necessary to be cost-effective. This examination depended on the electricity price value given by the National Electric Power Company for medium-sized industrial factories, which was 0.089 JD/kWh (0.13 $/kWh). This examination was in addition based on the value of the iron price provided by Jordan Steel Company (JS) for the average market price of Fe electrodes, which is around 630 JD/ton (888.34 $/ton).

In this study, sacrificial electrodes, electrical energy, and chemical costs were taken into consideration as main cost components in the calculation of the total *OPC* for solar-powered EC using the following equation [77–79]:

$$OPC = aENC + bELC + cCHC + SludgeCost \tag{10}$$

where *OPC* is the total operational cost of electrocoagulation (JD/m³ or $/m³), *CHC* is the consumption of neutralizing chemical alum (kg/m³), *a* is the electrical energy price (JD/kWh or $/kWh), and *b* and *c* are the electrodes and NaCl prices (JD/kg or $/kg), respectively.

It is worth mentioning that this analysis only applies to the aforementioned prices. If there is any change in the reference prices, significant changes may occur. Figure 11 and Table 7 show the total operational cost of conventional and solar EC treatment as a function of current density after 60 min of EC time.

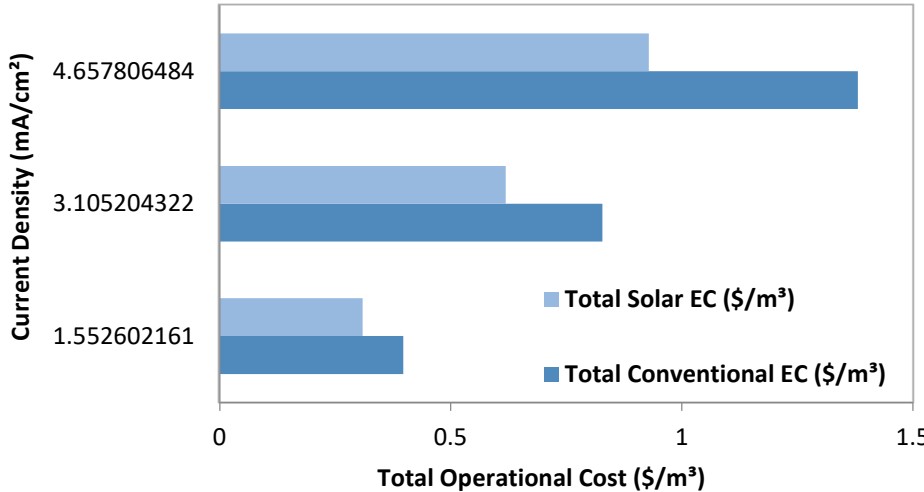

**Figure 11.** Total Operational Cost of Conventional and Solar EC as a Function of CD.

**Table 7.** Total operational cost (OPC) analysis for solar photovoltaic electrocoagulation systems using various current densities.

| Item | Unit | Current Density (mA/cm²) | | |
|---|---|---|---|---|
| | | **1.553** | **3.105** | **4.658** |
| Energy Consumption | kWh/m³ | 0.6 | 1.533 | 3.4 |
| Electrode Consumption | Fe kg/m³ | 0.3481 | 0.696 | 1.045 |
| Chemicals | kg/m³ | 0.0005 | 0.0005 | 0.0005 |
| Energy Cost | 0.13 $/kWh | 0.078 | 0.1993 | 0.442 |
| Electrode Cost | 0.89 $/kg | 0.31 | 0.619 | 0.93 |
| Chemical Cost | 20 $/kg | 0.01 | 0.01 | 0.01 |
| Total Conventional EC | $/m³ | 0.398 | 0.829 | 1.382 |
| Total Solar EC | $/m³ | 0.31 | 0.619 | 0.93 |

As shown in Figure 11 and Table 7, the operating cost of solar electrocoagulation increases largely as the CD increases (from 1.553 to 4.658 mA/cm²). It was in addition found that the lowest operating cost was 0.31 $/m³ at a current density of 1.553 mA/cm², and the corresponding removal efficiency was 80.4%. However, because we employed a solar photovoltaic cell for this purpose, we did not have to pay for the *ENC*. This means that the total cost is reduced by 22.1% (0.088 $/m³). The results of the EC treatment in terms of operating costs are consistent with those of Al Qedra [80], which is a treatment to remove boron from seawater using SPEC.

Table 8 presents the estimated values of energy and electrode consumption and the operating cost. The operation cost was estimated at treatment times of 5, 10, 20, 30, 40, 50, and 60 min. Figure 12 shows the relationship between operating cost and reaction time at different voltages and electrical currents.

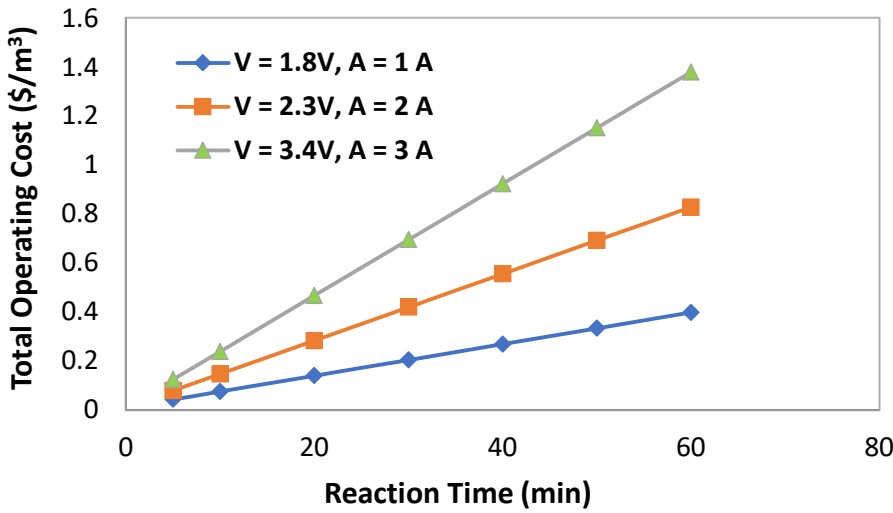

**Figure 12.** The operating cost for EC process with reaction time.

**Table 8.** Estimated values of electrode and energy consumption, and total conventional operating cost EC.

| Time (min) | V = 1.8 (V) CD = 1.553 (mA/cm²) | | | V = 2.3 (V) CD = 3.105 (mA/cm²) | | | V = 3.4 (V) CD = 4.658 (mA/cm²) | | |
|---|---|---|---|---|---|---|---|---|---|
| | ENC kWh/m³ | ELC kg/m³ | OPC $/m³ | ENC kWh/m³ | ELC kg/m³ | OPC $/m³ | ENC kWh/m³ | ELC kg/m³ | OPC $/m³ |
| 5 | 0.05 | 0.029 | 0.042 | 0.128 | 0.058 | 0.078 | 0.283 | 0.087 | 0.124 |
| 10 | 0.1 | 0.058 | 0.075 | 0.256 | 0.116 | 0.147 | 0.567 | 0.174 | 0.239 |
| 20 | 0.2 | 0.116 | 0.139 | 0.511 | 0.232 | 0.283 | 1.133 | 0.348 | 0.467 |
| 30 | 0.3 | 0.174 | 0.204 | 0.767 | 0.348 | 0.420 | 1.7 | 0.522 | 0.696 |
| 40 | 0.4 | 0.232 | 0.269 | 1.022 | 0.464 | 0.556 | 2.267 | 0.696 | 0.924 |
| 50 | 0.5 | 0.290 | 0.333 | 1.278 | 0.580 | 0.693 | 2.833 | 0.870 | 1.153 |
| 60 | 0.6 | 0.348 | 0.398 | 1.533 | 0.696 | 0.829 | 3.4 | 1.045 | 1.382 |

Figure 12 and Table 8 show that the operating cost increases with increasing the reaction time, electrical current, and voltage. The total cost at maximum COD removal efficiency (94.4%) at a voltage of 2.3 V and a current of 2A was 0.283 $/m³. However, because we employed a solar photovoltaic cell for this purpose, we did not have to pay for the *ENC*. This means that the total cost is reduced by 27% (0.076 $/m³) at the optimum conditions.

## 5. Conclusions

The purpose of this study was to investigate the feasibility of treating pharmaceutical wastewater with combined CC using alum and solar-powered EC with iron electrodes. The influence of the various operational parameters on the removal of COD has been studied. The following conclusions may be drawn from the findings of this study:

a.  The COD removal efficiency is increased by decreasing the current density, number of electrodes, and distance between electrodes. Meanwhile, it increased with the alum dose and reaction time.

b.  First- and second-order kinetic models were investigated on the EC. The first-order kinetic model was shown to be more suitable than the second-order kinetic model, with (higher $R^2$) values.

c.  Photovoltaic energy sources have shown to be more efficient and thus more economically feasible than conventional energy sources.

    d.   Finally, the study results showed that a combination of EC and CC processes in pharmaceutical wastewater treatment proved effective for the removal of COD.

## 6. Recommendations

The main recommendations arising from this study for future researchers are:

a.   The results of the combined CC and EC processes in this research may motivate researchers to adopt combination treatment methods since they show that such systems can produce water that is suitable for reuse in agriculture and irrigation.

b.   An important parameter in the EC process is the type of electrodes used. This issue needs more investigation. The most commonly used types are Al and Fe electrodes. Al electrodes have shown higher removal efficiencies than Fe. However, it is more expensive, and it produces sludge that needs special management.

c.   More studies should be conducted to investigate and optimize the most efficient electrode arrangement.

d.   The use of kinetic models to describe the treatment processes in these combined systems is still very limited. For this reason, it is necessary to develop suitable models for these new systems. If these models precisely describe the experimental results, they can be used in the scaling up of these systems [59].

e.   According to our findings, the combined system had a removal effectiveness of 94.4%. This encourages researchers to apply this integrated system to more contaminated industrial wastewater.

f.   $H_2$ production and conversion into electrical energy to reduce overall energy consumption.

g   The application of a sustainable treatment process in which the recovery of valuable materials in the wastewater should be performed before or after the treatment process [81].

**Author Contributions:** Conceptualization, T.M.A.-Z. and A.A.-J.; Methodology, Z.A.-Q.; Formal analysis, T.M.A.-Z. and A.A.-J.; Investigation, Z.A.-Q. and T.M.A.-Z.; Resources, A.A.-J.; Data curation, T.M.A.-Z.; Writing—original draft, T.M.A.-Z.; Writing—review & editing, Z.A.-Q. All authors have read and agreed to the published version of the manuscript.

**Funding:** This research received no external funding.

**Data Availability Statement:** No more data is available.

**Conflicts of Interest:** The authors declare no conflict of interest.

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
