# Peer review of "Performance, Modeling, and Cost Analysis of Chemical Coagulation-Assisted Solar Powered Electrocoagulation Treatment System for Pharmaceutical Wastewater"

_water, doi:10.3390/w15050980_

Round 1

Reviewer 1 Report

Authors focus on the Chemical Coagulation Assisted Electrocoagulation for the Treatment of Pharmaceutical Wastewater. I did not find any novelty of the study and presentation is poor. The removal of pharmaceutical is important area of research. 

1. Authors need to discuss the novelty of the present study. 

2. Authors need to mention importance of the Chemical Coagulation Assisted Electrocoagulation. 

3. Fig. 3 presentation should be simple... its like massy.. I suggest more simplest way to present. 

4. I suggest revised all figures with similar font size.

5. Incorporate comparative table for comparing removal efficiency. 

6. Authors should discuss the mechanism of removal in details.

Author Response

The response is attached

Reviewer 2 Report

-The title looks like a review paper. Revise with an attractive research title. 

-Graphical abstract: revise the word "Pharmaceutical wastewater source". "Chemical coagulation" Add some technical points, - What is the novelty of this research work? Already so much research works done on this combination. Apart from that how is it new? -Introduction: add permissible standards. Add specific importance to pharmaceutical wastewater, -Line 79-84: Elaborate the text with other combined processes. -Materials and methods: add details about the sample storage process. -Materials section, add a list of materials and the company name where you purchased them. -Site the COD procedure. Also, discuss the experimental procedure. -BOD experimental part and method were not discussed in the materials session, revise it. -Pharmaceutical wastewater contains Nitrate interference during COD analysis. The author can discuss that very clearly. -Figure-2: Add apparatus names directly. No need for numbers. Also, avoid scientific space losses. -2.2.2: revise the title. -Line 130. Revise the sentence. -Differentiate the Anode and Cathode. Also, mention clearly why Fe mesh or iron sheet or iron rod. Which model do you use for degradation? -Discuss the mechanism as you mentioned in Formula 1,2 and 3 inside the text in detail. -Add recent year references.  -Abstract and conclusion are almost similar. Revise it.

Author Response

The response is attached

Reviewer 3 Report

Specific Comments

1.      Graphical abstract should be revised. Increase the resolution.

2.      Introductions need to be revised. The objective should be strong and novel findings. The introduction part needs to add more recent references.

3.      Figure quality should be increased. Also, the Font size is not clear. Please check all the figures.

4.      Fig.2. Font size not clear.

5.      Fig.3. X and Y- axis units use brackets. X-axis should be Coagulant Dosage (mg/L) and Y-axis should be COD Removal (%). The author please check all the figures for such kind of mistakes.

6.      All figure's clarity should be improved, and legends are very small and should be improved.

7.      Table 2. (1st order and 2nd order) should replace by the First order kinetics model and Second order kinetics model.

8.      Figure 8 & 9. move to supplementary. Figure 9 needs to add R2 value.

9.      Figure 10. Need more discussion.

10.  Grammar and typos. The manuscript contains some grammatical and typographical errors. The authors need to thoroughly revise the manuscript and correct the errors.

11.  Moreover, there are a few corrections that need to be adopted for the overall presentation of this manuscript.

12.  Graphical abstract should be solid and novelty.

Author Response

The response is attached

Round 2

Reviewer 1 Report

Accept

Reviewer 2 Report

There is no further questions from my side

Reviewer 3 Report

The authors have provided detailed responses to each comment and query raised by reviewers. I am satisfied with the comments. They have incorporated all the changes in the revised manuscript.